# The Activity of Matrix Metalloproteinases (MMP-2, MMP-9) and Their Tissue Inhibitors (TIMP-1, TIMP-3) in the Cerebral Cortex and Hippocampus in Experimental Acanthamoebiasis

**DOI:** 10.3390/ijms19124128

**Published:** 2018-12-19

**Authors:** Natalia Łanocha-Arendarczyk, Irena Baranowska-Bosiacka, Izabela Gutowska, Agnieszka Kolasa-Wołosiuk, Karolina Kot, Aleksandra Łanocha, Emilia Metryka, Barbara Wiszniewska, Dariusz Chlubek, Danuta Kosik-Bogacka

**Affiliations:** 1Department of Biology and Medical Parasitology, Pomeranian Medical University in Szczecin, 70-204 Szczecin, Poland; nlanocha@pum.edu.pl (N.Ł.-A.); kotkarolina17@gmail.com (K.K.); 2Department of Biochemistry and Medical Chemistry, Pomeranian Medical University in Szczecin, 70-204 Szczecin, Poland; ika@pum.edu.pl (I.B.-B.); emilia_metryka@o2.pl (E.M.); dchlubek@pum.edu.pl (D.C.); 3Department of Biochemistry and Human Nutrition, Pomeranian Medical University in Szczecin, 70-204 Szczecin, Poland; izagut@poczta.onet.pl; 4Department of Histology and Embryology, Pomeranian Medical University in Szczecin, 70-204 Szczecin, Poland; Agnieszka.Kolasa@pum.edu.pl (A.K.-W.); kzhe@pum.edu.pl (B.W.); 5Department of Haematology and Transplantology, Pomeranian Medical University in Szczecin, 70-204 Szczecin, Poland; aleksandra.lanocha@pum.edu.pl

**Keywords:** matrix metalloproteinases (MMP), tissue inhibitor of metalloproteinases (TIMP), *Acanthamoeba* spp., cerebral cortex, hippocampus

## Abstract

The pathological process occurring within the central nervous system (CNS) as a result of the infection by *Acanthamoeba* spp. is not fully understood. Therefore, the aim of this study was to determine whether *Acanthamoeba* spp. may affect the levels of matrix metalloproteinases (MMP-2,-9), their tissue inhibitors (TIMP-1,-3) and MMP-9/TIMP-1, MMP-2/TIMP-3 ratios in the cerebral cortex and hippocampus, in relation to the host’s immunological status. Our results showed that *Acanthamoeba* spp. infection can change the levels of MMP and TIMP in the CNS and may be amenable targets for limiting amoebic encephalitis. The increase in the activity of matrix metalloproteinases during acanthamoebiasis may be primarily the result of inflammation process, probably an increased activity of proteolytic processes, but also (to a lesser extent) a defense mechanism preventing the processes of neurodegeneration.

## 1. Introduction

*Acanthamoeba* spp. can penetrate into the human central nervous system (CNS) and cause granulomatous amoebic encephalitis (GAE), a chronic infection often confused with bacterial or viral invasions, with non-specific symptoms and mortality exceeding 95% [1,2,3]. Trophozoites of these amoebae usually reach the CNS through the bloodstream from the site of primary pathology, such as the cornea of the eye, skin (ulceration), and the respiratory system (lungs) [4,5,6]. After the blood–brain barrier (BBB) has been compromised by the parasites, neutrophils and macrophages release mediators of inflammatory reactions as well as reactive oxygen species and nitric oxide [7]. The inflammation results in the production of cytokines, including tumor necrosis factor (TNF) and interleukins (IL-1, IL-6), leading to a synergistic effect on endothelial cells, inducing the synthesis of adhesive particles [2,8,9]. The pathological process occurring within the CNS as a result of the infection by *Acanthamoeba* spp. is not fully understood. In vitro studies show that increased permeability of the BBB is enhanced by extracellular serine proteases which degrade the tight junction proteins [2]. One of the factors responsible for the breaching of the blood–brain barrier, the degeneration of myelin proteins, as well as those acting on cytokines and chemokines, are extracellular matrix metalloproteinases (MMP) [10]. Metalloproteinases-2 (MMP-2) and -9 (MMP-9) regulate growth, proliferation, cell apoptosis, degrade type IV collagen, and have the potential to damage the basement membrane [11,12]. Their role has been demonstrated in inflammatory and infectious diseases of the nervous system in experiments on laboratory animals, e.g., in bacterial meningitis. The induction of their activity allows the lymphocytes to migrate through the compromised BBB and contribute to nerve tissue damage [13]. An increased activity of MMP-2 and MMP-9 has been observed in infections of tropical protozoa to the CNS, e.g., *Plasmodium falciparum* and *Trypanosoma brucei gambiense* [14]. In addition, the expression of some MMP increases in opportunistic infections, including *Toxoplasma gondii*, *Leishmania* spp. and *Trypanosoma cruzi* in immunocompromised hosts [15]. Lam et al. [16] suggested that MMP plays a critical role in the in vivo infection of the CNS, and that amenable targets may exist for limiting *Naegleria fowleri* brain infection. The cellular mechanism underlying *Acanthamoeba* spp. brain infection in relation to MMP and tissue inhibitors of metalloproteinases (TIMP) in experimental acanthamoebiasis in relation to the host immunological status is largely unknown. TIMP provide the necessary balance to prevent excessive degradation of extracellular matrix molecules. Disturbances in the balance between the metalloproteinases and their tissue inhibitors (MMP/TIMP) are most often associated with progressive pathological changes in the nervous system [17]. A strong correlation observed between MMP and TIMP may suggest a possible role of immune mediators in the immunopathogenesis of viral brain infections [18]. The role of MMP-2 and MMP-9 and their tissue inhibitors TIMP-1 and TIMP-3 in extracellular proteolysis is directly connected with the formation and maintenance of a normal perineuronal network structure surrounding neurons, which is important for the creation of new synaptic connections. As no studies to date have addressed the implications of the role of MMP and TIMP in *Acanthamoeba* spp. brain infections in immunocompetent or immunosuppressed hosts, the purpose of this study was to determine whether *Acanthamoeba* spp. may affect the levels of MMP (-2,-9), their tissue inhibitors TIMP (-1,-3) and MMP-9/TIMP-1, MMP-2/TIMP-3 ratios in the cerebral cortex and hippocampus, in relation to the host’s immunological status.

## 2. Results 

### 2.1. MMP-2 in the Cerebral Cortex and Hippocampus during Acanthamoebiasis

The highest levels of MMP-2 in *Acanthamoeba* spp. infected immunocompetent mice were found in the cerebral cortex at 8 days post *Acanthamoeba* spp. infection (dpi) (0.22 ng/mg protein) and the lowest in the hippocampus in control immunocompetent mice at 24 dpi (0.07 ng/mg protein). MMP-2 in the cerebral cortex of the immunosuppressed had a significantly higher level compared to immunocompetent *Acanthamoeba* spp. infected mice at 16 dpi (Figure 1). 

### 2.2. MMP-9 in the Cerebral Cortex and Hippocampus during Acanthamoebiasis

The highest level of MMP-9 was demonstrated in the hippocampus of *Acanthamoeba* spp. infected immunocompetent mice at 8 dpi (651.67 pg/mg protein) and the lowest in the cerebral cortex of control immunocompetent mice at 8 dpi (137.17 pg/mg protein). In the *Acanthamoeba* spp. infected immunocompetent mice the level of MMP-9 protein in the cerebral cortex was significantly higher than in the control group at 8 dpi (by 72%, *p* = 0.02). In the cerebral cortex of *Acanthamoeba* spp. infected mice we found downregulation of MMP-9 levels at 16 dpi, but the observed differences were not statistically significant (Figure 2). We noted a significantly lower MMP-9 level in the hippocampus at 24 dpi in the immunosuppressed *Acanthamoeba* spp. infected mice (AS group) compared to the immunosuppressed uninfected mice (CS group) (*p* = 0.04). Moreover, MMP-9 level in the *Acanthamoeba* spp. infected immunocompetent mice differed significantly between the cerebral cortex and hippocampus at 16 dpi (*p* = 0.03). There were no significant differences in MMP-9 levels in the studied brain structures between immunocompetent and immunosuppressed *Acanthamoeba* spp. infected mice.

### 2.3. TIMP-1 in the Cerebral Cortex and Hippocampus during Acanthamoebiasis 

In the *Acanthamoeba* spp. infected immunocompetent mice the level of TIMP-1 protein in the cerebral cortex was significantly higher than in the control group at 8 dpi (by 80.3%, *p* = 0.04). There was significant difference in TIMP-1 level in the cerebral cortex between the immunocompetent and immunosuppressed *Acanthamoeba* spp. infected mice at 8 dpi (by 61%, *p* = 0.01). We found significant downregulation of TIMP-1 protein in the hippocampus of the *Acanthamoeba* spp. infected immunocompetent mice compared to the immunosuppressed infected group (by 62.7%, *p* = 0.05) at 8 dpi. There were significant differences in TIMP-1 levels in the cerebral cortex and hippocampus between the immunocompetent and immunosuppressed *Acanthamoeba* spp. infected mice (by 41.6%, *p* = 0.04) at 8 dpi and (by 73.2%, *p* = 0.02) at 16 dpi (Figure 3). 

### 2.4. TIMP-3 in the Cerebral Cortex and Hippocampus during Acanthamoebiasis 

The level of TIMP-3 in the cerebral cortex was significantly higher in the immunocompetent than immunosuppressed *Acanthamoeba* spp. infected mice at 8 dpi (by 81.7%, *p* = 0.008) (Figure 4). We observed downregulation in the hippocampus of immunosuppressed *Acanthamoeba* spp. infected mice compared to the immunosuppressed control group at 16 dpi (by 69%, *p* = 0.04). TIMP-3 level in *Acanthamoeba* spp. infected immunosuppressed mice in the cerebral cortex did not exceed 0.2 ng/mg. There were no significant differences in TIMP-3 level between the cerebral cortex and hippocampus.

### 2.5. MMP-9/TIMP-1 and MMP-2/TIMP-3 Ratios in the Brain Structures 

The MMP-9/TIMP-1 ratio was greater in the hippocampus of the immunosuppressed *Acanthamoeba* spp. infected mice (AS group) than of the control at the beginning of infection. A similar trend was noted in *Acanthamoeba* spp. infected immunosuppressed mice in MMP-2/TIMP-3 ratio at 8 dpi, but the difference was not significant.

### 2.6. Immunohistochemistry

#### 2.6.1. MMP-2

In neocortex MMP-2 immunoexpression (Figure 5a) was mainly detected at the beginning of the experiment (8 dpi) in all studied group of mice. In infected animals (Figure 5a(A,D)) it was more prominent than in control (Figure 5a(G,J)). During the following day of infection/suppression, in the IHC analysis the reactivity was almost absent (Figure 5a(B,C, E,F, H,I, K,L)). In the hippocampus, mainly neurons of granular cell layer (GCL) of dentate gyrus (Figure 5b) and pyramidal cell layer (PyrCL) of cornu ammonis (Figure 5c) exhibited immunoexpression of detected proteins. Therefore, these cells were taken into consideration in the discussion of MMP-2 immunolocalization in the hippocampus. In dentate gyrus, similarly MMP-2 occurred mainly at the beginning of experiment (8 dpi) in all studied group of mice. However, it was more evident and spread across the entire GCL in immunocompetent mice (Figure 5b(A,G); red asterisks); immunosuppressed animals showed occasionally visible MMP-2-positive neurons in the granular cell layer (Figure 5b(D,J)). Generally, at 16 dpi and 24 dpi of all group immunoreactivity was weak and rare, almost no detectible (Figure 5b(B,C,E,F,H,I,K,L)), such as in neocortex. Similarly to neocortex and dentate gyrus, in cornu ammonis MMP-2 (Figure 5c(A–L)) could be observed mainly at the beginning of experiment (8 dpi) in the studied groups of mice (A, AS and C). Generally, at 16 and 24 dpi immunoreactivity in all groups was weak and rare, almost undetectable (Figure 5c(B,C,F,I,K,L)). 

#### 2.6.2. MMP-9

The results of the immunohistochemical reactions (IHC) showed that in the neocortex (Figure 6a), dentate gyrus (Figure 6b) and cornu ammonis (Figure 6c) of mice from control immunocompetent group of mice (C; Figure 6a–c(G–I)) the intensity of immunoexpression of MMP-9 was different from MMP-9 expression in immunosuppressed uninfected control group (CS, Figure 6a–c(J–L)), immunosuppressed infected with *Acanthamoeba* spp. (AS, Figure 6a–c(D–F)) and immunocompetent infected mice (A, Figure 6a–c(A–C)). The observed changes depended on the duration of infection or immunosuppressant treatment. The results of IHC reactions in different part of murine brains (Figure 6a–c) reflected the results of ELISA analysis and also showed the fluctuating intensity tensions of MMP-9 immunoexpression. In neocortex (Figure 6a) MMP-9 immunoexpression was more pronounced in animals infected by *Acanthamoeba* (Figure 6a(A–F)) than in controls (Figure 6a(G–L)) especially at the initial (Figure 6a(A,D)) and final/decay (Figure 6a(C–F)) stages of infection. However, in the CS group the intensity of MMP-9 immunoreactivity was also high at 24 dpi (Figure 6a(L)). Immunoexpression of MMP-9 in dentate gyrus (Figure 6b) of immunosuppressed infected (Figure 6b(D–F)) and control mice (Figure 6b(G–I)) was higher than in immunocompetent infected (Figure 6b(A–C)) or immunosuppressed non-infected mice (Figure 6b(J–L)). Generally in the hippocampus of mice with increased immunoreactivity to MMP-9, we observed a tendency to increased immunoexpression of the studied marker, especially in the basal region of GCL (Figure 6b, white asterisk). The highest MMP9-immunoreactivity is noticeable in pyramidal cell layer (PyrCL) of *cornu ammonis* of infected mice in the course of all infection (Figure 6c(A–C)) and also high in 8 postinfection day of immunosuppressed mice (Figure 6c, D); control groups of mice the immunosuppressant treatment result in rise of MMMP9 immunoexpression (Figure 6c(J–L)) in comparison to control, untreated animals (Figure 6c(G–I)). 

#### 2.6.3. TIMP-1

TIMP-1 immunoexpression in neocortex (Figure 7a) was more intense in mice infected with *Acanthamoeba* spp. (Figure 7a(A–F)) than in uninfected animals (Figure 7a(G–L)). In the infected group immunoreactivity was most intense at 8 dpi (Figure 7a(A)) then at the following days of the experiment (Figure 7a(B,C)), and opposite to immunosuppressed infected mice, was lower at 8 dpi (Figure 7a(D)), higher at 16 and 24 dpi (Figure 7a(E–F)). Similarly, in immunocompetent control (Figure 7a(G–I)) and immunosuppressed control groups (Figure 7a(J,L)) the TIMP-1 immunoexpression was a little bit higher during the following days of experiment (16 and 24 dpi). Immunohistochemical procedure showed TIMP-1 expression in dentate gyrus of hippocampus mainly in the *Acanthamoeba* spp. infected mice (Figure 7b(A–C)) and in the immunosuppressed uninfected animals (Figure 7b(J–L)) over the entire period of the experiment. The expression of TIMP-1 in AS (Figure 7b(D–F)) and in C (Figure 7b(G–I)) groups of mice was rather low. In cornu ammonis, the nerve cells immunoreactivity for TIMP-1 (Figure 7c) was much higher in immunocompetent infected mice (Figure 7c(A–C)) than in control animals (Figure 7c(G–I)); the immunesuppressive treatment caused abolition of the immunoreactivity in both groups of mice (Figure 7c(D–F,J–L)).

#### 2.6.4. TIMP-3

The lowest level of the TIMP-3 (Figure 8a) was observed in the neocortex of control mice (Figure 8a(G–I)); the *Acanthamoeba* spp. infection (Figure 8a(A–C)) and (Figure 8a(D–F))/or (Figure 8a(J–L)) immunosuppressive treatment resulted in enhancement/intensification of the immunoreactivity, regardless of the duration of the experiment. In the dentate gyrus, the highest expression of TIMP-3 (Figure 8b) was visible in the infected animal at the beginning of infection (Figure 8b(A)) and then decreased (Figure 8b(B,C)); in the control group the expression of TIMP-3 was similar during the entire period of experiment (Figure 8b(G–I)), immunosuppression decreased TIMP-3 expression (Figure 8b(D–F,J–L)) in comparison to controls, i.e., immunocompetent *Acanthamoeba* spp. infected (A group) and immunocompetent uninfected (C group) mice. In the cornu ammonis of the hippocampus, the intensity of immunoexpression of TIMP-3 (Figure 8c(A–L)) was very similar to that observed in dentate gyrus (Figure 8b(A–L))–the highest in the immunocompetent infected mice (Figure 8c(A–C)), high in control (Figure 8c(G–I)) and the lowest in immunosuppressed infected/ uninfected mice (Figure 8c(D–F,J–L)). 

## 3. Discussion

Some opportunistic parasitic invasions are accompanied by the imbalance between MMP and endogenous MMP inhibitors, mostly in favor of active proteolysis [14]. The inflammation induced by parasites may be associated with the influx of leukocytes, among others to the brain, resulting in immunopathology and collateral tissue damage [15]. In the case of parasitic cerebral infections, MMP activity is crucial for the migration of inflammatory cells and parasites as well as BBB integrity disorders [18]. It is noted that *Acanthamoeba* spp. and *Balamuthia mandrillaris*, GAE etiologic agents, invade the host organism by ingesting tissue and producing metalloproteinase enzymes [19]. Similarly, *Naegleria fowleri* trophozoites, isolated from a fatal case of primary amebic meningoencephalitis, secreted MMP that play a role in the brain infection [16]. Immunosuppression appears to be a factor in brain infections caused by *Acanthamoeba* spp., which are predominantly lethal and often diagnosed postmortem [20].

MMP-2 and MMP-9 play a role in neurological disorders. MMP-2 is constitutively expressed by several cell types including brain tissue, and participates in BBB damage and immunological pathogenesis [12]. In addition, it plays a role in the growth, infiltration and formation of cancer metastases [21]. Similar to monocytes, T-cells, astrocytes, microglia, and macrophages, neutrophils flow to damaged tissues and release large amounts of MMP-9 from their granules [14,22]. 

This study demonstrated that in the *Acanthamoeba* spp. infected immunocompetent mice the level of MMP-9 protein in the cerebral cortex was significantly higher than in the control group at the beginning of infection. Moreover, we noted that MMP-9 level in the amoeba infected immunocompetent hosts was significantly different between brain structures at 16 dpi. Some researchers found that MMP-2 and MMP-9 are expressed in the hippocampus, striatum, diencephalus, mesencephalon, frontal cortex, and cerebellum of rats, with the activity of both proteins being the highest within the hippocampus, the brain part responsible for the conversion of short-term to long-term memory via long-term potentiation (LTP) [23]. Our study confirmed the variability of MMP concentrations within brain structures in acanthamoebiasis in both immunocompetent and immunocompromised hosts. 

We also found a significant upregulation of MMP-2 levels in the cerebral cortex of *Acanthamoeba* spp. infected immunosuppressed mice compared to amoeba–infected immunocompetent hosts, but only at 16 dpi. In some studies, the increasing levels of MMP-2 and MMP-9 correlated with the presence of parasites and leukocytes in the cerebrospinal fluid (CSF) of *Trypanosoma brucei gambiense* infected patients with the final stage of African sleeping sickness [14]. Jacintho et al. [24] noted increasing levels of MMP-9 in dogs with visceral leishmaniasis. Also patients with Chagas heart disease complicated by heart failure had higher serum levels of MMP-9 [25]. In experimental *Trypanosoma cruzi* infection, Gutierrez et al. [26] showed that the inhibition of MMP-2 and MMP-9 increased inflammation.

The secretion of MMP is initiated by various factors, e.g., platelet-derived growth factor (PDGF), interleukin-1 (IL-1), tumor necrosis factor alpha (TNF-α) and phagocytosis. The secretion is inhibited by transforming growth factor β (TGF-β) and glucocorticoids. During *Toxoplasma gondii*–induced immune response and inflammation, IL-1, IL-23, TNF-α, and cyclooxygenase-2 (COX-2) stimulated the production of MMP in the brain [27].

In this study we found that at 24 dpi, the level of MMP-9 was significantly lower in the hippocampus of the *Acanthamoeba* spp. infected mice with reduced immunity, induced by methylprednisolone (drug used to suppress the immune system and decrease inflammation). Corticosteroids have been shown to suppress the expression of MMP-9 in CSF during acute CNS inflammation [28]. Synthetic MMP inhibitors (including steroids) may inhibit the production of prostaglandin 2 (PGE_2_) and cyclic AMP (cAMP) that also mediate the production of MMP. Green et al. [29] found that dexamethasone decreased the cerebrospinal fluid MMP-9 concentrations early in the treatment and this may represent one mechanism by which corticosteroids improve outcomes in meningitis.

MMPs participate in neurodegeneration, with recent research also indicating their involvement in the development of nervous system and neuroplastic processes activated as a result of nerve tissue damage [12]. Changes in the concentration of MMP-2 and MMP-9 are independent of each other, and both of these enzymes may play different roles in damage and regeneration of nervous system tissue. Increased MMP-2 synthesis in the brains of mice infected with *Acanthamoeba* spp. may be associated with regenerative processes that occur due to the initiation of CNS damage by this parasite. Costanzo and Perrino [30] observed that an olfactory nerve interruption caused an increase in the concentration of MMP-9 as well as upregulation of MMP-2 during recovery. The level of MMP-2 changed independently of MMP-9, probably due to the significant contribution of this metalloproteinase in axonal regeneration. Due to the fact that MMP activity during brain acanthamoebiasis is important for the migration of inflammatory cells and parasites as well as BBB integrity disorders, these enzymes might represent suitable therapeutic targets to prevent breaching of the brain–blood barrier in amoebic brain infection.

Interactions between MMP and TIMP can play a significant role in parasite pathogenesis based on the mechanisms of BBB substrate degradation and also function as effectors and regulators of the immune response [14]. TIMP-1 and TIMP-3 are capable of inhibiting various proteases and TNF-α [31]. TIMP upregulation in infected tissues does not compensate for the increased activity of proteolytic enzymes. Some researchers suggest that TIMP-1 may inhibit pathogen clearance during infection by limiting MMP-driven lymphocyte penetration into the CNS [14]. In animal experimental models of CNS this MMP inhibitor plays an important role in neuroprotection, neural plasticity and tissue repair [31]. MMP inhibitor production may increase to create a normal MMP/TIMP ratio and to regain BBB integrity by decreasing activated T cell migration [32].

This study demonstrated that TIMP-1 and TIMP-3 increased in the cerebral cortex in *Acanthamoeba* spp. infected immunocompetent mice at the beginning of infection. There was also a significant increase in TIMP-1 in the hippocampus of *Acanthamoeba* spp. infected immunocompetent mice compared to the cerebral cortex of *Acanthamoeba* spp. infected immunosuppressed mice. TIMP-3 levels in the cerebral cortex were significantly higher in immunocompetent than immunosuppressed *Acanthamoeba* spp. infected mice at 8 dpi. Moreover, infection with *Acanthamoeba* spp. in the hippocampus did not affect TIMP-3 in immunocompetent mice; a significant decrease in this inhibitor was observed only in immunosuppressed *Acanthamoeba* spp. infected mice at 16 dpi.

The aforementioned variable activity of synthesized TIMP-1 and TIMP-3 inhibitors during brain acanthamoebiasis may be related to their different form of appearance in brain tissues. TIMP-1 is produced in a soluble form and can be degraded more quickly, while TIMP-3 is insoluble, bound to the intercellular matrix, and may decompose at a slower rate [33]. In addition, the decrease in the amount of TIMP may be due to increased MMP catalytic activity, and conversely, an increase in inhibitors may indicate a reduction in MMP activity. TIMP can act independently of MMP, involved in the regulation of cellular processes, including cell proliferation and apoptosis, and angiogenesis [34].

TIMP investigated in this study can participate in the regulation of inflammation and the cell cycle. TIMP-3 inhibits not only metalloproteinases but limits the activity of adamalysin-like proteinases (ADAM), including ADAM-17—an enzyme converting TNF-α (TACE /ADAM 17) [35]. TNF-α can stimulate phagocytosis, proliferation of T and B lymphocytes, increase natural killer (NK) cell toxicity and stimulate the expression of MMP [36]. It is possible that an elevated concentration of TIMP-3 in the cerebral cortex and hippocampus of *Acanthamoeba* spp. infected immunocompetent mice may result from ongoing inflammation in the brain. Similarly, TIMP-1 and TIMP-3 mRNA expression was noted to be upregulated in *T. cruzi* infected heart tissue [26]. Some authors suggest that TIMP-1 and TIMP-3 may contribute to pathology in Post-kala-azar dermal leishmaniasis (PKDL) and may inhibit wound healing [37].

Due to the different roles of matrix metalloproteinases and their specific inhibitors, we analyzed the coefficients assessing the relationships between them. We found that MMP-9/TIMP-1 and MMP-2/TIMP-3 ratios were low and similar between the groups and brain structures, but increased in the hippocampus in *Acanthamoeba* spp. infected immunosuppressed mice (AS) at the beginning of infection. It is possible that in the AS group in the acute phase of the infection, the proteolytic activity of all components of the gelatinase inhibitor system was stimulated. Chronic immune activation and inflammation as well as immunodeficiency cause a homeostasis imbalance between MMP and TIMP in HIV positive patients, especially in those with neurocognitive disorders [15,38]. Lichtinghagen et al. [39] also observed the upregulation of MMP-9/TIMP-1 ratio in patients with multiple sclerosis.

## 4. Materials and Methods

### 4.1. Experimental Animal Model

The experimental course of acanthamoebiasis under study was described in detail in our earlier works [40,41]. The study had been approved by the Local Ethics Committee for Experiments on Animals in Szczecin (No. 29/2015, dated 22 June 2015) and Poznań (No. 64/2016 dated 09 September 2016). It was conducted on 96 male Balb/c mice obtained from a licensed breeder (the Center of Experimental Medicine, Medical University in Białystok, Poland). At the start of the experiment, the age of the mice was 6–10 weeks. The animals had genetic and health certificates issued by a veterinarian. All animals were weighed, and their mean weight was 23 g. The animals were also evaluated by clinical observations, housed singly on a 12-h light/dark cycle, and fed Labofeed H feed (Morawski, Kcynia, Poland) and water *ad libitum*.

Inoculation of the mice was performed using *Acanthamoeba* spp. strain AM 22, isolated from a patient with acute myeloid leukemia (AML) and atypical pneumonia [42]. Parasites were grown in NN Agar covered with a suspension of deactivated *Escherichia coli* and incubated at 37 °C according to standard methods [43]. Strain AM 22 not only has pneumophilic properties, as we observed in our previous study [40], but also neurophilic effects, where numerous amoeba trophozoites were re-isolated from brain fragments from *Acanthamoeba* spp. infected immunocompetent and immunosuppressed mice.

For the in vivo studies animals were divided into 4 groups:immunocompetent control group uninfected mice (C, *n* = 18),immunocompetent *Acanthamoeba* spp. infected mice (A, *n* = 30),immunosuppressed *Acanthamoeba* spp. infected mice (AS, *n* = 30),immunosuppressed uninfected mice (CS, *n* = 18).

Animals were immunosuppressed by administering 0.22 mg (10 mg/kg) methylprednisolone sodium succinate (MPS, Solu-Medrol, Pfizer, Europe MA EEIG; cat. no.: W07908) in 0.1 mL of 0.9% saline intraperitoneally (i.p.) at −4, −3, −2, −1 and 0 days before inoculation with the amoeba. The model of immunosuppression in experimental acanthamoebiasis using MPS was based on literature data [44].

The mice from groups A and AS were inoculated intra-nasally with 3 μL of a suspension containing 10–20 thousand amoebae. Control animals (C and CS groups) were given the same volume of sterile solution (3 μL of 0.9% NaCl solution). Euthanasia of the *Acanthamoeba* spp. infected mice was conducted at 8, 16, and 24 days post infection (dpi), depending on the clinical signs and degree of infection, with a peritoneal overdose of sodium pentobarbital (Euthasol vet, FATRO) (2 mL/kg body weight) and subsequently necropsied. The virulence of the amoebae was determined by the degree of infection. Fragments (5 × 5 mm) of the brain were placed on NN agar and incubated at 41 °C to assess infection intensity. The cerebral cortex and hippocampus samples were fixed and stored in 4% buffered formalin solution (Avantor, Poland; cat. no.:432173111) for histological analyses and fixed in liquid nitrogen, and then stored at −80 °C biochemical analyses.

### 4.2. Determination of MMP-2, MMP-9, TIMP-3, TIMP-1 in the Brain

The collected brain structures (hippocampus, cerebral cortex) were homogenized in a hammer mill in liquid nitrogen, then lysis buffer was added to the tissue powder, followed by homogenization in a knife homogenizer. In order to perform the lysis of tissues and release relevant epitopes, the samples were subjected to five thaw-freeze cycles, then centrifuged (3000 G × 20 min × 4 °C) and the obtained supernatant used for further determinations.

The concentrations of MMP and TIMP were determined with commercially available quantitative ELISAs: MMP-2 (Mouse Matrix Metalloproteinase 2 ELISA Kit), MyBioSource (San Diego, CA, USA), cat. no.: MBS454416; MMP-9 (Mouse Matrix Metalloproteinase 9 ELISA Kit), MyBioSource, cat. no.: MBS355322; TIMP 3 (Mouse Tissue Inhibitors of Metalloproteinase 3 ELISA Kit) (San Diego, CA, USA) cat. no.: 454748; TIMP 1 (Mouse Tissue Inhibitors of Metalloproteinase 1 ELISA Kit) (San Diego, CA, USA) cat. no.: 2881293. The determinations were made according to the instructions provided by the manufacturers using an ASYS UVA 340 spectrophotometer.

### 4.3. Measurement of Total Protein Concentration by BCA

In order to standardize the obtained results, the samples were assayed for total protein level with bicinchoninic acid (BCA) using a commercially available kit (Pierce™ BCA Protein Assay Kit; (Thermofisher Waltham, MA, USA); cat. no.: 23225). The spectrometric measurement was carried out on an ASYS UVA 340 spectrophotometer.

### 4.4. I.mmunolocation of MMP-2, MMP-9, TIMP-1 and TIMP-3 by Immunohistochemical Methods

Paraffin-embedded sections (3–5 μm) of mouse brains were immunostained for the visualization of matrix metalloproteinases (MMP) and their inhibitors (TIMP) expression. Immunohistochemistry was performed using specific primary antibodies, monoclonal against MMP-2 (Thermofisher; Waltham, MA, USA) cat. no.: 436000; final dilution—1:100), MMP-9 (Thermofisher; Waltham, MA, USA) cat. no.: MA5-15886; final dilution—1:500), TIMP-1 (Thermofisher; Waltham, MA, USA) cat. no.: MA5-13688; final dilution—1:100) and polyclonal against TIMP-3 (Thermofisher; Waltham, MA, USA) cat. no.: PA5-26133; final dilution—1:25). First, the deparaffinized sections were microwave irradiated in citrate buffer (pH 6.0) to heat induce epitope retrieval. After slow cooling to room temperature, slides were washed in phosphate-buffered saline (PBS) solution twice for 5 min and then incubated with primary antibodies over the night in 4 °C. Following this, sections were stained using an avidin-biotin-peroxidase system with diaminobenzidine (DAKO LSAB+System-HRP; DakoCytomation, Glostrup, Denmark, cat. no.: K0679) as the chromogen, in conformity with the staining procedure instructions included. Sections were washed in distilled H_2_O and counterstained with hematoxylin. For a negative control, specimens were processed in the absence of primary antibodies. Positive staining was defined microscopically (Leica DM5000 B, Wetzlar, Germany) by visual identification of brown pigmentation.

### 4.5. Statistical Analysis

The obtained results were analyzed statistically using Statistica 13.1 software. Arithmetical means and standard deviations (SD) were calculated for each of the studied parameters. In order to assess differences between the parameters studied, Kruskal-Wallis ANOVA followed by Mann–Whitney-U tests were used. Correlations between the parameters were examined with Spearman’s rank correlation coefficient (r_s_). Differences were considered statistically significant at *p* < 0.05.

## 5. Conclusions

Our results showed that *Acanthamoeba* spp. infection can change the levels of matrix metalloproteinases and tissue inhibitor of metalloproteinase in the central nervous system. The increase in the activity of matrix metalloproteinases during acanthamoebiasis may be primarily the result of inflammation process, probably an increased activity of proteolytic processes, but also (to a lesser extent) a defense mechanism preventing the processes of neurodegeneration. In addition, the level of both MMP and TIMP in brain tissue are up- or downregulated in *Acanthamoeba* spp. mice depending on the immunological status of the host and the brain structure studied. As MMP activity during brain acanthamoebiasis is important for the migration of inflammatory cells and parasites and BBB integrity disorders, these enzymes might represent suitable therapeutic targets to prevent the unsealing of brain–blood barrier in inducted amoebic brain infection.

## Figures and Tables

**Figure 1 ijms-19-04128-f001:**
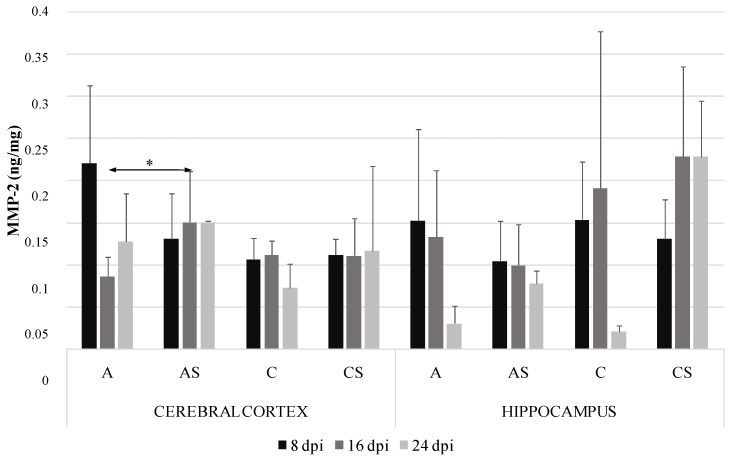
Matrix metalloproteinases-2 (MMP-2) level (ng/mg protein) in the cerebral cortex and hippocampus in control and infected groups at 8, 16 and 24 days post *Acanthamoeba* spp. infection (dpi). Data represent means ± SD for 6 independent experiments. C, immunocompetent uninfected control group mice; CS, immunosuppressed uninfected control group mice; A, immunocompetent *Acanthamoeba* spp. infected mice; AS, immunosuppressed *Acanthamoeba* spp. infected mice; * *p* ≤ 0.05 for the significance of difference (Mann–Whitney U test).

**Figure 2 ijms-19-04128-f002:**
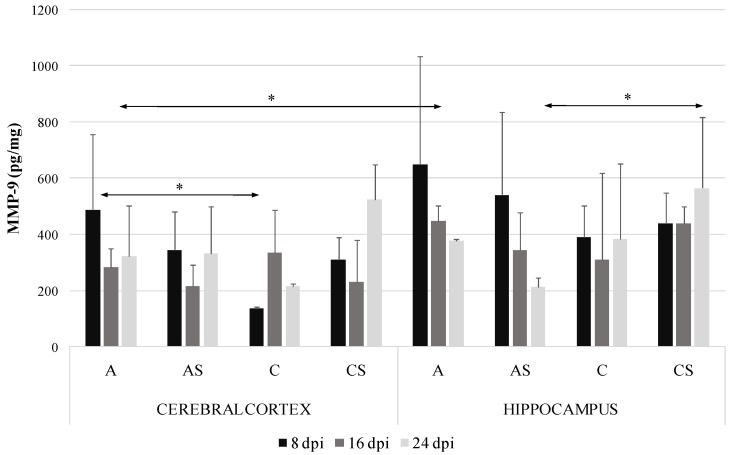
Matrix metalloproteinases-9 (MMP-9) level (pg/mg protein) in the cerebral cortex and hippocampus in control and infected groups at 8, 16 and 24 days post *Acanthamoeba* spp. infection (dpi). Data represent means ± SD for 6 independent experiments. C, immunocompetent uninfected control group mice; CS, immunosuppressed uninfected control group mice; A, immunocompetent *Acanthamoeba* spp. infected mice; AS, immunosuppressed *Acanthamoeba* spp. infected mice; * *p* ≤ 0.05 for the significance of difference (Mann–Whitney U test).

**Figure 3 ijms-19-04128-f003:**
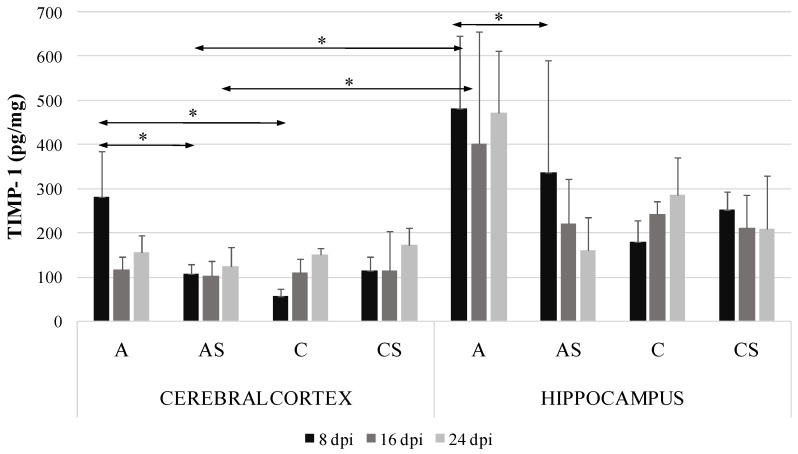
Tissue inhibitors of metalloproteinase-1 (TIMP-1) level (pg/mg protein) in the cerebral cortex and hippocampus in control and infected groups at 8, 16 and 24 days post *Acanthamoeba* spp. infection (dpi). Data represent means ± SD for 6 independent experiments. C, immunocompetent uninfected control group mice; CS, immunosuppressed uninfected control group mice; A, immunocompetent *Acanthamoeba* spp. infected mice; AS, immunosuppressed *Acanthamoeba* spp. infected mice; * *p* ≤ 0.05 for the significance of difference (Mann–Whitney U test).

**Figure 4 ijms-19-04128-f004:**
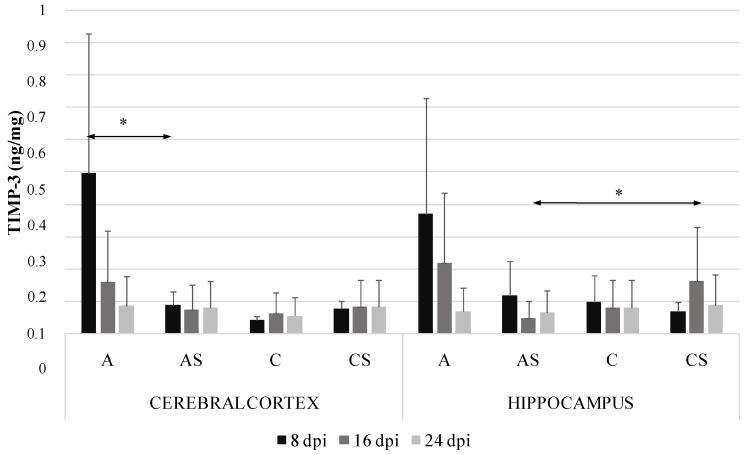
Tissue inhibitors of metalloproteinase-3 (TIMP-3) level (ng/mg) in the cerebral cortex and hippocampus in control and infected groups at 8, 16 and 24 days post *Acanthamoeba* spp. infection (dpi). Data represent means ± SD for 6 independent experiments. C, immunocompetent uninfected control group mice; CS, immunosuppressed uninfected control group mice; A, immunocompetent *Acanthamoeba* spp. infected mice; AS, immunosuppressed *Acanthamoeba* spp. infected mice; * *p* ≤ 0.05 for the significance of difference (Mann–Whitney U test).

**Figure 5 ijms-19-04128-f005:**
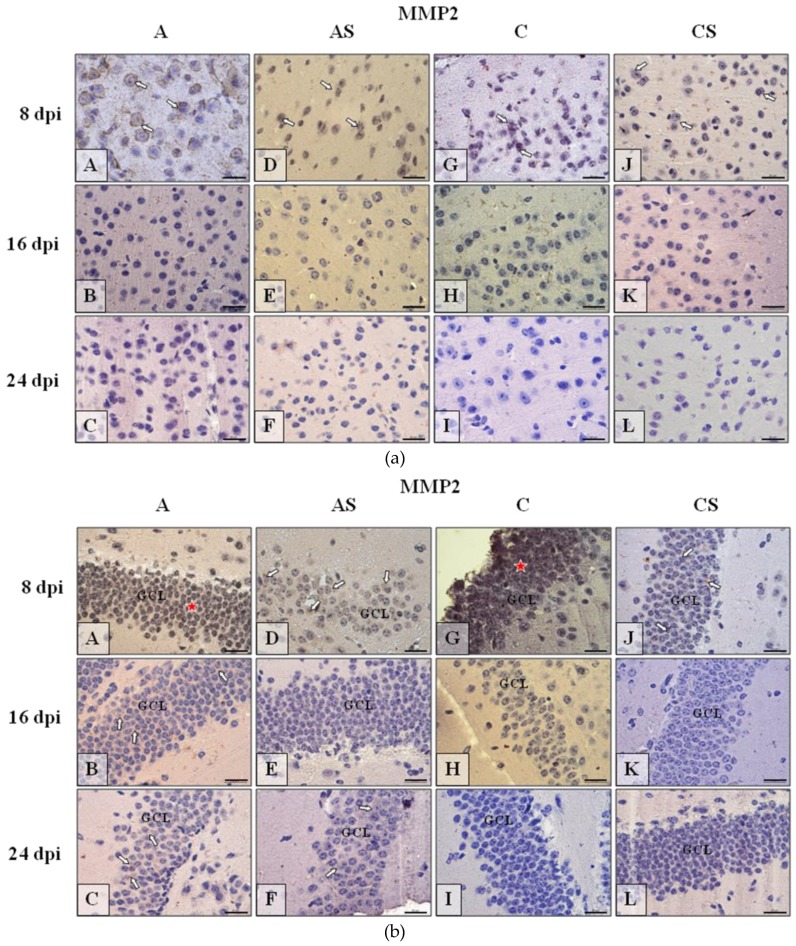
(**a**) Representative microphotography showing immunoexpression (brown color) of matrix metalloproteinases-2 (MMP-2) in the neocortex of mice from control group (a, G–I), immunosuppressed uninfected control group (a, J–L), immunocompetent infected by *Acanthamoeba* spp. (a, A–C) and immunosuppressed *Acanthamoeba* spp. infected group (a, D–F) in 8, 16, and 24 day post infection (dpi). White arrows show immunoexpression of MMP-2 in the neurons of cerebral cortex at the beginning of experiment (5a, A,D,G,J); in following days of experiment (5a, B,C,E,F,H,I,K,L) the MMP-2 in IHC procedure is under detection level. (**b**) Representative microphotography showing immunoexpression (brown color) of MMP-2 in neurons of granular cell layer (GCL) of the gyrus dentatus of mice from control group (b, G–I), immunosuppressed uninfected control group (b, J–L), immunocompetent infected by *Acanthamoeba* spp. (b, A–C) and immunosuppressed *Acanthamoeba* spp. infected group (b, D–F) in 8, 16, and 24 dpi. White arrows show immunoexpression of MMP-2 in the neurons of that region of hippocampus. Red asterisks, in some cases (b, A,G), shows tendency to immunolocalization of MMP-2 over the whole thickness of GCL. In following days of experiment (b, B,C,E,F,H,I,K,L) the MMP-2 in IHC procedure is under detection level. (**c**) Representative microphotography showing immunoexpression (brown color) of MMP-2 in the neurons of pyramidal cell layer (PyrCL) of cornu ammonis of mice from control group (c, G–I), immunosuppressed uninfected control group (c, J–L), immunocompetent infected by *Acanthamoeba* spp. (c, A–C) and immunosuppressed *Acanthamoeba* spp. infected group (c, D–F) in 8, 16, and 24 dpi. White arrows show immunoexpression of MMP-2 in the neurons of hippocampus mainly at the beginning of experiment (c, A,D,G); in following days of experiment (c, B,C, F, H,I, J–L) the MMP-2 in IHC procedure is under detection level. Objective magnification a–c: ×100. Scale bar = 20 μm.

**Figure 6 ijms-19-04128-f006:**
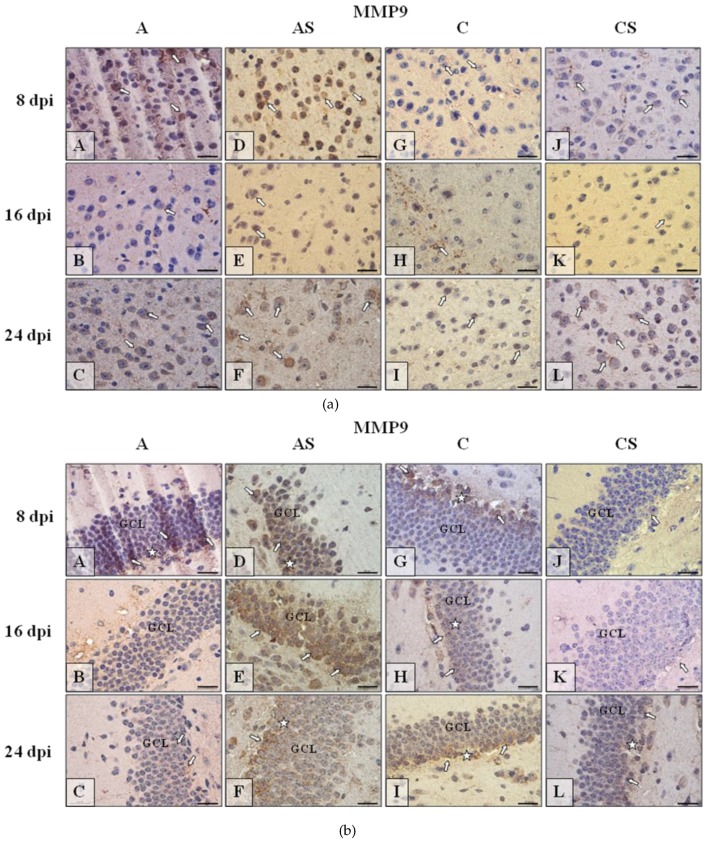
(**a**) Representative microphotography showing immunoexpression (brown color) of matrix metalloproteinases-9 (MMP-9) in the neocortex of mice from control group (a, G–I), immunosuppressed uninfected control group (a, J–L), immunocompetent infected by *Acanthamoeba* spp. (a, A–C) and immunosuppressed *Acanthamoeba* spp. infected group (a, D–F) in 8, 16 and 24 day post infection (dpi). White arrows show immunoexpression of MMP9 in the neurons of cerebral cortex. (**b**) Representative microphotography showing immunoexpression (brown color) of MMP-9 in the neurons of granular cell layer (GCL) of gyrus dentatus of mice from control group (6b, G–I), immunosuppressed uninfected control group (b, J–L), immunocompetent infected by *Acanthamoeba* spp. (b, A–C) and immunosuppressed *Acanthamoeba* spp. infected group (b, D–F) in 8, 16, and 24 dpi. White arrows show immunoexpression of MMP-9 in the neurons of that region of hippocampus. White asterisk, in some cases, shows tendency to immunolocalization of MMP-9 in basal region of GCL. **c)** Representative microphotography showing immunoexpression (brown color) of MMP-9 in neurons of pyramidal cell layer (PyrCL) of cornu ammonis of mice from control group (c, G–I), immunosuppressed uninfected control group (c, J–L), immunocompetent infected by *Acanthamoeba* spp. (c, A–C) and immunosuppressed *Acanthamoeba* spp. infected group (c, D–F) in 8, 16, and 24 dpi. White arrows show immunoexpression of MMP-9 in neurons of that region of hippocampus. Objective magnification **a**–**c**: ×100. Scale bar = 20 μm.

**Figure 7 ijms-19-04128-f007:**
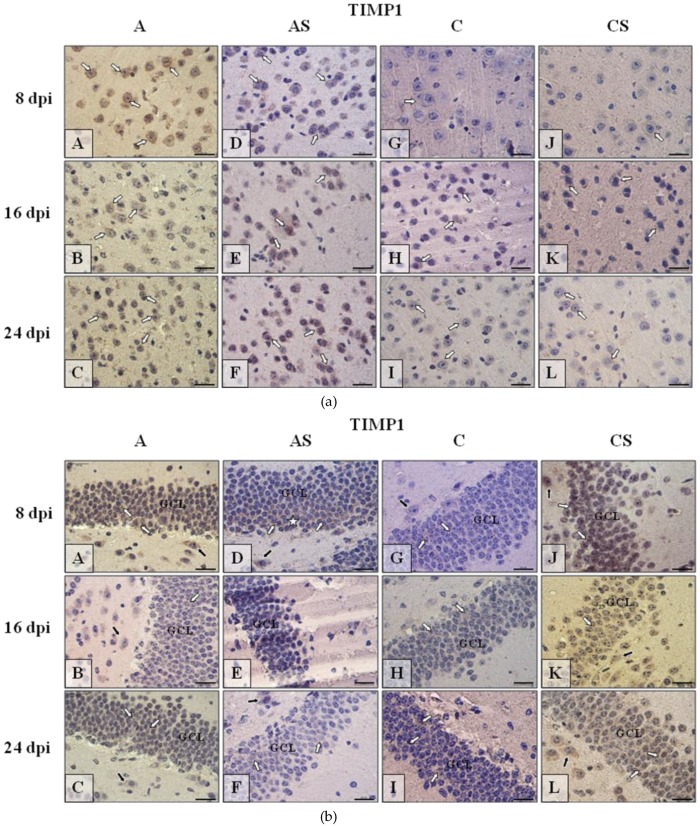
(**a**) Representative microphotography showing immunoexpression (brown color) of tissue inhibitors of metalloproteinase-1 (TIMP-1) in the neocortex of mice from control group (a, G–I), immunosuppressed uninfected control group (a, J–L), immunocompetent infected by *Acanthamoeba* spp. (a, A–C) and immunosuppressed *Acanthamoeba* spp. infected group (a, D–F) in 8, 16, and 24 day post infection (dpi). White arrows show immunoexpression of TIMP-1 in neurons of cerebral cortex. (**b**) Representative microphotography showing immunoexpression (brown color) of TIMP-1 in neurons of granular cell layer (GCL) of the gyrus dentatus of mice from control group (b, G–I), immunosuppressed uninfected control group (b, J–L), immunocompetent infected by *Acanthamoeba* spp. (b A–C) and immunosuppressed *Acanthamoeba* spp. infected group (b, D–F) in 8, 16, and 24 dpi. White arrows show immunoexpression of TIMP-1 in neurons of GCL of hippocampus, black arrows show immunoreactivity of big neurons of polymorphic cell layer. White asterisk (b, D) shows tendency to immunolocalization of TIMP-1 in basal region of GCL. (**c**) Representative microphotography showing immunoexpression (brown color) of TIMP-1 in neurons of pyramidal cell layer (PyrCL) of the cornu ammonis of mice from control group (c, G–I), immunosuppressed uninfected control group (c, J–L), immunocompetent infected by *Acanthamoeba* spp. (c, A–C) and immunosuppressed *Acanthamoeba* spp. infected group (c, D–F) in 8, 16, and 24 dpi. White arrows show immunoexpression of TIMP-1 in neurons of that region of hippocampus. Objective magnification **a**–**c**: ×100. Scale bar = 20 μm.

**Figure 8 ijms-19-04128-f008:**
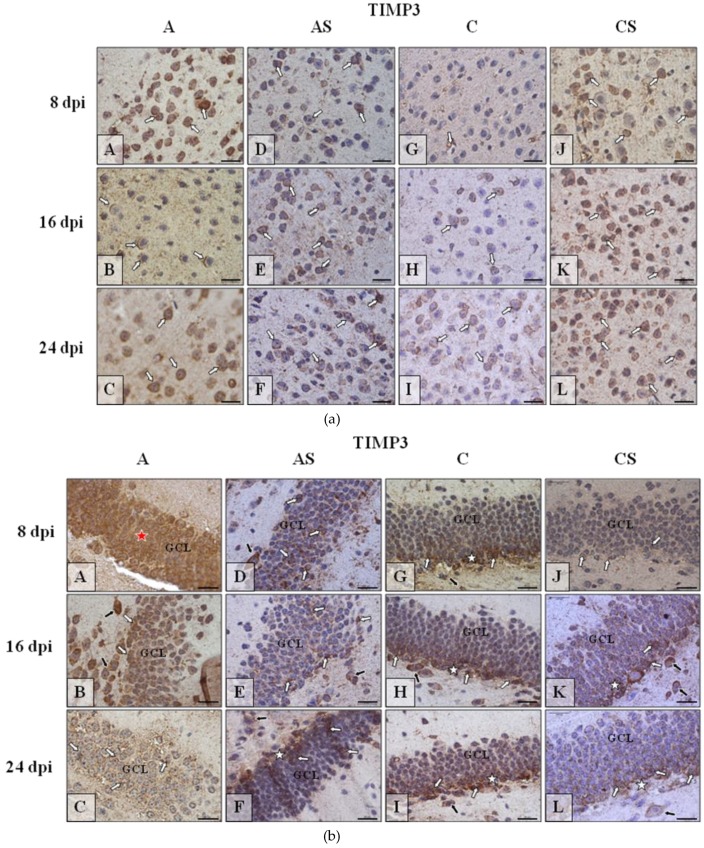
(**a**) Representative microphotography showing immunoexpression (brown color) of tissue inhibitors of metalloproteinase-3 (TIMP-3) in the neocortex of mice from control group (a, G–I), immunosuppressed uninfected control group (a, J–L), immunocompetent infected by *Acanthamoeba* spp. (a, A–C) and immunosuppressed *Acanthamoeba* spp. infected group (a, D–F) in 8, 16, and 24 day post infection (dpi). White arrows show immunoexpression of TIMP-3 in neurons of cerebral cortex. (**b**) Representative microphotography showing immunoexpression (brown color) of TIMP-3 in neurons of granular cell layer (GCL) of the gyrus dentatus of mice from control group (b, G–I), immunosuppressed uninfected control group (b, J–L), immunocompetent infected by *Acanthamoeba* spp. (b, A–C) and immunosuppressed *Acanthamoeba* spp. infected group (b, D–F) in 8, 16, and 24 dpi. White arrows show immunoexpression of TIMP-3 in neurons of GCL of hippocampus, black arrows show immunoreactivity of big neurons of polymorphic cell layer. Red asterisk (A) shows tendency to immunolocalization of TIMP-3 over the whole thickness of GCL. (**c**) Representative microphotography showing immunoexpression (brown color) of TIMP-3 in the neurons of pyramidal cell layer (PyrCL) of cornu ammonis of mice from control group (c, G–I), immunosuppressed uninfected control group (c, J–L), immunocompetent infected by *Acanthamoeba* spp. (c, A–C) and immunosuppressed *Acanthamoeba* spp. infected group (c, D-F) in 8, 16, and 24 dpi. White arrows show immunoexpression of TIMP-3 in neurons of that region of hippocampus. Red asterisks, in some cases (c, A,G–J), shows tendency to immunolocalization of TIMP3 over the whole thickness of GCL.Objective magnification **a**–**c**: ×100. Scale bar = 20 μm.

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
