# Peer review of "The Activity of Matrix Metalloproteinases (MMP-2, MMP-9) and Their Tissue Inhibitors (TIMP-1, TIMP-3) in the Cerebral Cortex and Hippocampus in Experimental Acanthamoebiasis"

_ijms, 2018, doi:10.3390/ijms19124128_

Round 1
Reviewer 1 Report
The manuscript provided by Kosik-Bogacka Danuta demonstrated that Acanthamoeba spp. infection can change the levels of MMP and TIMP in the CNS, which could provided a potential target for amoebic encephalitis. The author devoted great efforts to measure the MMP-2/9 and TIMP-1-3 in the cerebral cortex and hippocampus during acanthamoebiasis. They also showed the corresponding IHC results to further characterize their findings. Overall, the manuscript was well organized and it was recommended to be published in IJMS if the author can address the following comments:
Major comments:
1. The authors cited a lot of references to describe the possible mechanism even in their conclusions, it is better if the author can provide some extra data to demonstrate the mechanism about their findings.
2. The conclusions part looks like a copy of the abstract except changing the abbreviations, it needs to be reorganized.
Minor comments
1. Line 159. Type error, “then in control” should be “than in control”
2. Line 261. “much more higher” should be “much higher”
3. Line 472. “fixed in in”, delete “in”
4. Line 584. “Virusdisease” should be italic
Author Response
Thank you very much for very thorough review. All the comments from the Reviewers have been addressed in the text of our manuscript.
1. The authors cited a lot of references to describe the possible mechanism even in their conclusions, it is better if the author can provide some extra data to demonstrate the mechanism about their findings.
Corrected according to the Reviewer's suggestion.
2. The conclusions part looks like a copy of the abstract except changing the abbreviations, it needs to be reorganized.
Corrected according to the Reviewer's suggestion.
Minor comments
1. Line 159. Type error, “then in control” should be “than in control”
Corrected according to the Reviewer's suggestion.
2. Line 261. “much more higher” should be “much higher”
Corrected according to the Reviewer's suggestion.
3. Line 472. “fixed in in”, delete “in”
Corrected according to the Reviewer's suggestion.
4. Line 584. “Virusdisease” should be italic
Corrected according to the Reviewer's suggestion.
Kind regards
Danuta Kosik-Bogacka
Reviewer 2 Report
The submitted manuscript, 'The activity of matrix metalloproteinases (MMP-2, MMP-9) and their tissue inhibitors (TIMP-1,TIMP-3) in the cerebral cortex and hippocampus in experimental acanthamoebiasis' discusses the role of metalloproteinases MMP-2 and 9 in Acanthamoebiasis. Authors have investigated the background studies well and provided ample evidences of MMP involvements in CNS infections. This study in particular has been planned and presented in well enough manner. I would suggest acceptance of this manuscript for publication in the journal.
I would suggest authors to provide catalog numbers of items/reagents used in this study to make their experimental procedures easily reproducible.
Author Response
Thank you very much for very thorough review. All the comments from the Reviewers have been addressed in the text of our manuscript.
The submitted manuscript, 'The activity of matrix metalloproteinases (MMP-2, MMP-9) and their tissue inhibitors (TIMP-1,TIMP-3) in the cerebral cortex and hippocampus in experimental acanthamoebiasis' discusses the role of metalloproteinases MMP-2 and 9 in Acanthamoebiasis. Authors have investigated the background studies well and provided ample evidences of MMP involvements in CNS infections. This study in particular has been planned and presented in well enough manner. I would suggest acceptance of this manuscript for publication in the journal.
I would suggest authors to provide catalog numbers of items/reagents used in this study to make their experimental procedures easily reproducible
Corrected according to the Reviewer's suggestion.
Kind regards
Danuta Kosik-Bogacka